# Genome-Wide Identification and Expression Analysis of the *Fructose-1,6-Bisphosphate Aldolase* (*FBA*) Gene Family in Sweet Potato and Its Two Diploid Relatives

**DOI:** 10.3390/ijms26157348

**Published:** 2025-07-30

**Authors:** Zhicheng Jiang, Taifeng Du, Yuanyuan Zhou, Zhen Qin, Aixian Li, Qingmei Wang, Liming Zhang, Fuyun Hou

**Affiliations:** Crop Research Institute, Shandong Academy of Agricultural Sciences/Scientific Observing and Experimental Station of Tuber and Root Crops in Huang-Huai-Hai Region, Ministry of Agriculture and Rural Affairs, Jinan 250100, China; pyxjzc@163.com (Z.J.);

**Keywords:** sweet potato, *Ipomoea trifida*, *Ipomoea triloba*, Fructose-1,6-bisphosphate aldolase, storage root development, starch biosynthesis, abiotic stress, hormone crosstalk

## Abstract

Fructose-1,6-bisphosphate aldolase (*FBA*; EC 4.1.2.13) is a key enzyme in glycolysis and the Calvin cycle, which plays crucial roles in carbon allocation and plant growth. The *FBA* family genes (*FBA s*) have been identified in several plants. However, their presence and roles in sweet potato remain unexplored. In this study, a total of 20 *FBAs* were identified in sweet potato and its wild wild diploidrelatives, including seven in sweet potato (*Ipomoea batatas*, 2*n* = 6*x* = 90), seven in *I. trifida* (2*n* = 2*x* = 30), and six in *I. triloba* (2*n* = 2*x* = 30). Their protein physicochemical properties, chromosomal localization, phylogenetic relationship, gene structure, promoter *cis*-elements, and expression patterns were systematically analyzed. The conserved genes and protein structures suggest a high degree of functional conservation among *FBA* genes. *IbFBAs* may participate in storage root development and starch biosynthesis, especially *IbFBA1* and *IbFBA6*, which warrant further investigation as candidate genes. Additionally, the *FBAs* could respond to drought and salt stress. They are also implicated in hormone crosstalk, particularly with ABA and GA. This work provides valuable insights into the structure and function of *FBAs* and identifies candidate genes for improving yield, starch content, and abiotic stress tolerance in sweet potatoes.

## 1. Introduction

Sweet potato (*Ipomoea batatas* (L.) Lam., 2*n* = B_1_B_1_B_2_B_2_B_2_B_2_ = 6*x* = 90) is classified as an autohexaploid species within the Convolvulaceae family, *Ipomoea* Genus, and *Batatas* Section. It is recognized for its high adaptability and is utilized as food, an industrial material, and a bioenergy resource [1,2,3]. In recent years, to maintain national food security, sweet potato cultivation has decreased and shifted towards marginal lands and low-quality soils [4]. Therefore, the exploration of genetic resources to enhance the yield, quality, and stress resistance of sweet potatoes is deemed of great importance for sweet potato production.

Fructose-1,6-bisphosphate aldolase (*FBA*; EC 4.1.2.13) is a key enzyme that catalyzes the reversible conversion of fructose-1,6-bisphosphate (FBP) into dihydroxyacetone phosphate (DHAP) and glyceraldehyde-3-phosphate (G3P) [5,6]. *FBA* enzymes are classified into two classes, and higher plants primarily contain Class I *FBA* enzymes [7,8]. Class I *FBAs* typically include a typical conserved TIM barrel structural domain and function as homotetramers [9,10]. In higher plants, *FBA* is localized in both the cytoplasm and chloroplasts, where it plays distinct roles in glycolysis and the Calvin cycle, respectively [11]. Therefore, *FBA* plays a significant role in plant growth and development by influencing photosynthesis, carbon assimilation, and partitioning. In previous research, the changes in *FBA* activity influence the photosynthesis and carbon allocation in potato, tobacco, and tomato, ultimately impacting the growth of plants [12,13,14]. In Arabidopsis, mutation of *AtFBA3* limits the biosynthesis of amino acids in roots by disrupting the plastid glycolysis [15]. In tea tree oil, the expression levels of *CoFBA1* and *CoFBA3* genes are highly correlated with the content of tea tree oil [16]. Furthermore, numerous studies have indicated that the *FBA* genes are involved in responses to various stresses, such as salinity [17,18], drought [19], and temperature [14,20]. These findings highlight *FBA* genes as promising targets for genetic engineering to improve crop yield, quality, and stress tolerance.

Recent advances in genomics have enabled the identification and characterization of *FBA* gene families in various plant species, including eight in *Arabidopsis thaliana* [21], eight in *Solanum lycopersicum* [22], sixteen in *Nicotiana tabacum* [23], seventeen in cotton [24], nine in *Solanum tuberosum* [25], and five in *Cucumis sativus* [26]. Recently, the release of genome assemblies of cultivated sweet potato [27] and its two wild diploid relatives, *Ipomoea trifida* and *I. triloba* [28], has provided valuable genomic resources for the identification of the *FBA* gene family in sweet potato and its two wild diploid relatives. In this study, a total of 20 *FBA* genes were identified, including seven in *I. batatas*, seven in *I. trifida*, and six in *I. triloba*. Comprehensive analyses were performed on these *FBA* genes, including phylogenetic relationships, synteny mapping, gene structure evaluation, and conserved motif identification. In addition, their organ specificity and expression patterns related to storage root development, starch biosynthesis, abiotic stress, and hormone responses were examined using qRT-PCR or RNA-seq. This research demonstrates the potential of *FBA* genes in improving sweet potato yield, quality, and stress tolerance, providing a foundation for future research and crop improvement strategies.

## 2. Results

### 2.1. Identification of FBAs in Sweet Potato and Its Two Wild Diploid Relatives

In this study, three strategies (i.e., blastp search, hmmer search, and the CD-search database) were used to completely identify *FBAs* in sweet potato and its two wild diploid relatives, *I. trifida* and *I. triloba* genomes. We identified seven, seven, and six *FBAs* in *I. batatas*, *I. trifida*, and *I. triloba*, respectively. *FBAs* were designated as “Ib*FBA*1–Ib*FBA*7”, “Itf*FBA*1–Itf*FBA*7”, and “Itb*FBA*1–Itb*FBA*6” based on the chromosomal locations, respectively. Table 1 presented the specific details of *FBAs*, including the genome length (ranged from 1682 bp to 2902 bp), CDS length (ranged from 1047 bp to 1182 bp), the number of amino acid (ranged from 348 aa to 393 aa), protein molecular weight (ranged from 37.14 kDa to 42.49 kDa), and theoretical isoelectric point (*p*I, ranged from 6.86 to 8.61). The basic characteristics analysis showed that all Ib*FBAs* were predicted to be stable with an instability index of less than 40. The grand average of hydropathicity (GRAVY) values of Ib*FBAs* ranged from −0.336 to −0.13, indicating that they are hydrophilic proteins. Subcellular localization prediction showed that Ib*FBA*2, Ib*FBA*4, Ib*FBA*5, and Ib*FBA*6 were localized to the cytoplasm, while Ib*FBA*1, Ib*FBA*3, and Ib*FBA*7 were localized to the chloroplasts (Table 1).

In *I. trifida*, coding sequence (CDS) lengths spanned 1074 to 1194 bp, while genomic sequences measured between 1752 and 3179 bp. Putative protein lengths varied from 357 to 397 amino acids (aa), corresponding to molecular weights (MW) of 38.23–42.66 kDa and isoelectric points (*p*I) ranging from 6.38 to 8.61. Primary characterization revealed all Itf*FBA* proteins as stable, exhibiting instability indices below 40. Grand average of hydropathicity (GRAVY) values, falling between −0.336 and −0.094, confirmed their hydrophilic nature. Subcellular localization predicted Itf*FBA*2, Itf*FBA*4, Itf*FBA*5, and Itf*FBA*6 in the cytoplasm, whereas Itf*FBA*1, Itf*FBA*3, and Itf*FBA*7 were assigned to chloroplasts (Table 1). For *I. triloba*, CDS lengths extended from 1077 bp to 1194 bp, with genomic lengths varying more broadly from 1831 bp to 4627 bp. Deduced polypeptides comprised 358 to 397 aa, yielding MWs of 38.51–42.69 kDa and pIs between 6.97 and 8.61. All Itb*FBAs* were similarly predicted to be stable (instability index < 40). Negative GRAVY scores affirmed the hydrophilicity of these proteins. Localization analysis indicated cytoplasmic positioning for Itb*FBA*2, Itb*FBA*4, and Itb*FBA*6, while Itb*FBA*1, Itb*FBA*3, and Itb*FBA*5 were chloroplast-targeted (Table 1).

In genomes of *I. batatas*, *I. triloba*, and *I. trifida*, *FBAs* were located on six chromosomes similarly (Figure 1). In *I. batatas*, two *IbFBAs* were mapped to LG14 and one each to LG2, LG6, LG7, LG11, and LG13 (Figure 1A). In *I. trifida*, *FBAs* were detected on Chr01 (1), Chr02 (1), Chr03 (1), Chr04(1), Chr09 (2), and Chr15 (1), and *I. triloba* exhibited a similar distribution of *FBAs*, except with a single *FBA* on Chr09 (Figure 1A,C). In order to further infer the evolution of *FBAs*, we carried out a synteny analysis among *IbFBAs*, *ItfFBAs*, and *ItbFBAs*. The results indicated that all *ItfFBAs* and *ItbFBAs* had one orthologous gene of *IbFBAs* (Figure 1D). These results showed that *FBA* genes were conserved in the process of evolution from diploid to hexaploid.

### 2.2. Phylogenetic Relationship Analysis of FBAs in Sweet Potato and Its Two Wild Diploid Relatives

To investigate the evolutionary relationships of *FBAs* in *I. batatas*, *I. trifida*, *I. triloba*, *Arabidopsis thaliana*, and *Solanum tuberosum*, we constructed a phylogenetic tree for 37 *FBAs* of these six species (i.e., 7 in *I. batatas*, 7 in *I. trifida*, 6 in *I. triloba*, 8 in *A. thaliana*, and 9 in *S. tuberosum*). According to the evolutionary distance, all the *FBAs* were divided into four groups and distributed unevenly across the phylogenetic tree branches (Figure 2). The detailed distributions of *FBAs* were as follows (total: *I. batatas*, *I. trifida*, *I. triloba*, *A.thaliana*, and *S. tuberosum*): Group I (14:3; 3, 3, 3, and 2), Group II (6:1; 1, 1, 2, and 1), Group III (12:2; 2, 1, 2, and 5) and Group IV (5:1; 1, 1, 1, 1, and 1) (Figure 2; Appendix A). Notably, all Ib*FBAs* clustered closely with their orthologs from *I. triloba* and *I. trifida*.

### 2.3. Conserved Motif and Exon–Intron Structure Analysis of FBAs in Sweet Potato and Its Two Wild Diploid Relatives

MEME analysis identified ten conserved motifs shared across Ib*FBAs*, Itf*FBAs*, and Itb*FBAs* (Figure 3A and Appendix A). Exon–intron structure analysis of *IbFBAs*, *ItbFBAs*, and *ItfFBAs* revealed minor variations (Figure 3B). The exons numbers ranged from 3 (*IbFBA4*, *IbFBA5*, *IbFBA6*, *ItfFBA4*, *ItfFBA5*, *ItfFBA6*, *ItbFBA4*, *ItbFBA5*, and *ItbFBA6*) to 6 (*IbFBA1*, *IbFBA3*, *ItfFBA1*, *ItfFBA3*, *ItbFBA1*, and *ItbFBA3*). The genes in Groups III and IV included more exons than those in Groups I and II. Specifically, all homologous *FBAs* in sweet potato and its two wild diploid relatives shared the same number of exons. These results indicated that the *FBA* genes were highly conserved during the evolution of sweet potatoes.

### 2.4. Cis-Element Analysis in the Promoters of FBAs in Sweet Potato and Its Two Wild Diploid Relatives

To elucidate the transcriptional regulation of *FBAs*, the 2 kb upstream promoter sequences of these genes were analyzed. Diverse *cis*-elements were identified and classified into five functional categories according to their roles: core/binding, light-responsive, developmental, hormone-responsive, and abiotic/biotic stress-responsive elements (Figure 4). Core/binding elements, specifically the TATA-box and CAAT-box, were ubiquitously present across all *IbFBA* promoters. Promoters of *IbFBA2*, *IbFBA4*, *IbFBA5*, and *IbFBA6* additionally harbored multiple AT-TATA-boxes. Cis-acting light-response motifs, including Box 4, GT1-motif, G-box, TCT-motif, and AAGAA-motif, occurred frequently among *IbFBA* promoters. Development-associated elements were detected in all *IbFBA* promoters except *IbFBA5*. Furthermore, the *IbFBA* promoter regions contained numerous phytohormone-associated cis-motifs, encompassing abscisic acid (ABA)-responsive elements (ABRE, ABRE4, ABRE3a), gibberellin (GA)-responsive elements (GARE-motif), auxin (IAA)-responsive elements (TGA-element), and jasmonate (JA)-responsive elements (CGTCA-motif, TGACG-motif) (Figure 4). Most of the *IbFBAs* promoters contained abiotic/biotic elements, such as drought-responsive elements (MYB and MYC), antioxidant response elements (ARE and STRE), low-temperature-responsive elements (ERE and WRE3), and biotic-stress-responsive elements (as-1) (Figure 4). Promoters of *ItfFBAs* and *ItbFBAs* exhibited similar profiles of *cis*-elements (Figure 4). These findings collectively suggested *FBAs’* involvement in growth/development regulation, hormone crosstalk, and stress adaptation, and the functions of *FBA* genes were highly conserved during the evolution of sweet potatoes.

### 2.5. Expression Analysis of FBAs in Sweet Potato and Its Two Wild Diploid Relatives

#### 2.5.1. Expression Analysis in Various Organs

In cultivated sweet potato, we carried out qRT-PCR to calculate the expression of *IbFBAs* in leaf, stem, and storage roots. *IbFBA1* and *IbFBA6* were highly expressed in storage roots, while other *IbFBAs* were highly expressed in leaves (Figure 5A).

To investigate the expression profiles of *ItfFBAs* and *ItbFBAs*, transcriptomic data derived from six distinct organs (flower bud, flower, leaf, stem, root1, root2) were analyzed [28]. Within *I. trifida*, predominant expressions of *ItfFBA1*, *ItfFBA2*, and *ItfFBA6* occurred in flower buds. Leaves exhibited elevated transcript levels for *ItfFBA3*, *ItfFBA5*, and *ItfFBA7*. Stem tissue showed high expression of *ItfFBA6*, while root1 displayed significant expression of *ItfFBA1* and *ItfFBA6*. *ItfFBA4* expression was notably high in root2 (Figure 5B). Crucially, none of the *ItfFBAs* were detected in floral tissues. In *I. triloba*, flower buds contained high transcript abundance for *ItbFBA1*, *ItbFBA2*, *ItbFBA4*, and *ItbFBA6*. Leaf-specific expression characterized *ItbFBA3*, *ItbFBA4*, and *ItbFBA5*. Stems featured predominant *ItbFBA4* expression, and root2 showed significant expression of *ItbFBA1* and *ItbFBA4* (Figure 5C). Expression for all *ItbFBAs* was absent in both flowers and root1. Collectively, these results indicate that *FBA* genes exhibit diverse organ-specific expression patterns across sweet potato and its two diploid progenitor species.

#### 2.5.2. Expression Analysis at Different Developmental Stages of Storage Roots in Sweet Potato

To investigate the potential roles of *IbFBAs* in storage root development of sweet potato, the expression patterns of *IbFBAs* were analyzed using various developmental stages (FR, fibrous roots; DR, developing storage roots; MR, mature storage roots) in the cultivar Taizhong 6, as well as data from a precocious variety (Jishu 29) and a late maturing line (Jishu 25) at different developmental stages (32, 46, and 67 day after planting (DAP)) [29,30]. In Taizhong 6, *IbFBA1*, *IbFBA4*, and *IbFBA6* were highly expressed in DR and showed a low expression level in FR, while *IbFBA3* and *IbFBA7* showed the opposite expression pattern (Figure 6A). In Jishu 29 and Jishu 25, *IbFBA3* and *IbFBA6* were highly expressed in 46 DAP; *IbFBA1*, *IbFBA2*, and *IbFBA4* were highly expressed in 67 DAP (Figure 6B).

We further used qRT-PCR to determine the expressions of *IbFBAs* at different developmental stages of storage roots (i.e., 30, 45, 60, 75, 90, 105, and 120 DAP) in the high-starch variety Jishu25. As shown in Figure 6C, *IbFBA1* and *IbFBA2* were highly expressed at 60 DAP (initial thickening), while *IbFBA3*, *IbFBA4*, *IbFBA6*, and *IbFBA7* were highly expressed at 75 DAP (rapid thickening and initial starch accumulation stage) (Figure 6C). In addition, *IbFBA5* remained consistently low in expression throughout sweet potato development. These results suggested that individual *IbFBAs* may have distinct roles in development and in starch and sugar accumulation across various stages of sweet potato storage root formation.

#### 2.5.3. Expression Analysis in Sweet Potato Lines with Different Starch Contents

According to the previous results, *IbFBAs* may play significant roles in starch biosynthesis in sweet potatoes. A transcriptome analysis showed that *IbFBA2*, *IbFBA3*, and *IbFBA6* were highly expressed in the high-starch-content variety Jishu 25 (Figure 6B). To further investigate the functions of *IbFBAs* in the starch biosynthesis of sweet potato, we analyzed the expression of *IbFBAs* in high-starch varieties (Jishu 25 and Jishu 36), medium-starch varieties (Jishu 26 and Yanshu 25), and low-starch varieties (Shanshu 1 and Jishu 33). Overall, *IbFBAs* were highly expressed in the high-starch lines. In particular, *IbFBA2*, *IbFBA5*, and *IbFBA6* exhibited significantly higher expression levels in the high-starch lines compared to the medium-/low-starch lines. *IbFBA1*, *IbFBA3*, *IbFBA4*, and *IbFBA7* did not display a clear expression trend (Figure 7).

#### 2.5.4. Expression Analysis Under Drought and Salt Stresses

To explore the prospective functions of *IbFBAs* in abiotic stresses (drought and salt stress) responses, we made use of the qRT-PCR from Jishu 25 subjected to PEG6000 treatment and the RNA-seq data from a salt-sensitive variety (NL54) and a salt-tolerant line (Jishu 26) subjected to NaCl stress at 0 h, 0.5 h, 6 h, and 12 h, and the expression patterns of *IbFBAs* were analyzed [31,32]. Under PEG6000 stress, *IbFBA1*, *IbFBA4*, *IbFBA5*, *IbFBA6*, and *IbFBA7* were upregulated, while others were downregulated or did not show significant changes in Jishu 25 (Figure 8A). Under NaCl stress, *IbFBA2*, *IbFBA4*, and *IbFBA5* were upregulated in both Jishu 26 and NL54, whereas *IbFBA6* and *IbFBA7* were downregulated in both Jishu 26 and NL54, and others showed no significant trend (Figure 8B).

Next, RNA-seq data for *I. trifida* and *I. triloba* under mannitol and NaCl stress indicated that specific *FBA* genes (*ItfFBA1*, *ItfFBA6*; *ItbFBA1*, *ItbFBA4*, and *ItbFBA6*) were upregulated under drought and salt stress (Appendix A). Taken together, these results indicated that certain *FBAs* participated in drought- and salt-stress responses in sweet potato and its two wild diploid relatives.

#### 2.5.5. Expression Analysis in Response to Hormones

Next, we conducted a qRT-PCR to evaluate the expression levels of *IbFBAs* in sweet potato variety Jishu25 following treatments with ABA, GA3, and IAA to explore the potential functions of *IbFBAs* in the hormone signaling and crosstalk in sweet potato (Figure 9). Under ABA and GA3 treatments, all *IbFBAs* were upregulated (Figure 9A,B). Under ABA treatment, *IbFBA1*, *IbFBA2*, *IbFBA5*, and *IbFBA6* exhibited the highest expression at 3 h, *IbFBA4* at 6 h, and *IbFBA3* and *IbFBA7* at 48 h (Figure 9A). Under GA3 treatment, *IbFBA3* and *IbFBA7* were highest expressed at 3 h; *IbFBA1* and *IbFBA6* at 6 h; and *IbFBA2*, *IbFBA4*, and *IbFBA5* at 24 h after the treatment (Figure 9B). Under IAA treatment, *IbFBA4*, *IbFBA5*, and *IbFBA6* were upregulated, and the others were downregulated or unchanged (Figure 9C).

RNA-seq data analysis revealed the expression of *ItfFBAs* and *ItbFBAs* subjected to ABA (upregulated genes: *ItfFBA1*, *ItfFBA6*, *ItbFBA1*, *ItbFBA4*, and *ItbFBA6*; downregulated genes: *ItfFBA3*, *ItfFBA4*, *ItfFBA5*, *ItfFBA7*, *ItbFBA3*, and *ItbFBA5*), GA3 (upregulated genes: *ItfFBA1*, *ItfFBA3*, *ItfFBA6*, and *ItbFBA6*; downregulated genes: *ItfFBA4*, *ItfFBA5*, *ItfFBA7*, *ItbFBA1*, *ItbFBA3*, *ItbFBA4*, and *ItbFBA5*), and IAA (upregulated genes: *ItfFBA3*, *ItfFBA4*, *ItfFBA7*, *ItbFBA3*, and *ItbFBA5*; downregulated genes: *ItfFBA1*, *ItfFBA6*, *ItbFBA1*, *ItbFBA4*, and *ItbFBA6*) treatments (Appendix A) [28]. Therefore, *FBAs* exhibited distinct expression patterns in response to hormone treatments and may be involved in hormonal crosstalk in sweet potato and its diploid relatives.

## 3. Discussion

### 3.1. FBAs Were Conserved During the Evolution of Sweet Potato

In various plant species, the *FBA* gene family has been identified in *Arabidopsis thaliana* (eight) [21], *Solanum lycopersicum* (eight) [22], *Nicotiana tabacum* (sixteen) [23], cotton (seventeen) [24], *Solanum tuberosum* (nine) [25], and *Cucumis sativus* (five) [26]. In this research, we identified seven, seven, and six *FBAs* in *I. batatas*, *I. trifida*, and *I. triloba*, respectively. According to the chromosomal localization and phylogenetic relationships of *FBAs*, the homologs among *I. batatas*, *I. trifida*, and *I. triloba* were located on similar genomic positions (Figure 1 and Figure 2), suggesting that *FBAs* of sweet potato originated from its diploid ancestors, with *I. batatas* being more closely related to *I. trifida.*

Furthermore, ten conserved motifs were identified in all *FBAs* (Figure 3A). Gene exon/intron structures are typically conserved among homologous genes of a gene family [33]. In our study, all homologous *FBAs* had the same number of exons and introns in *I. batatas*, *I. trifida*, and *I. triloba*. *FBAs* in Group III and Group IV had more exons than those in Group I and Group II (Figure 3B). These results demonstrate that *FBAs*, as pivotal enzymes in the pentose phosphate cycle, glycolysis, and gluconeogenesis pathways, were conserved during the evolution of *I. batatas* [10,18,34]. Owing to the conserved gene and protein structures, *FBAs* likely play crucial roles in maintaining the carbohydrate metabolism to keep cellular functions in plants.

### 3.2. IbFBAs Are Involved in Storage Root Development and Starch Biosynthesis in Sweet Potato

As an important metabolic enzyme in photosynthesis and glycolysis/gluconeogenesis, fructose-1,6-bisphosphate aldolase (*FBA*) is an important enzyme involved in photosynthetic products and energy metabolism [15]. In potatoes, the inhibition of *FBA* activity decreased the sucrose and starch content in tubers [12]. In tobacco, overexpression of *Arabidopsis thaliana* plastid aldolase gene (*AtptAL*) promoted the photosynthetic rate and enhanced biomass yields of tobacco [13]. In rice, the mutation of *OsALD-Y* exhibited serious defects in chloroplast development and Chlorophyll accumulation, affecting the photosynthetic rate and sugar metabolism in leaves, which ultimately reduced the yield of rice [35]. In Arabidopsis, the mutation of *AtFBA3* disrupted plastid glycolytic metabolism in roots, thereby limiting the synthesis of essential compounds such as starch and amino acids [15]. Overexpression of *PeFBA6* in rice increased the accumulation of glucose, fructose, and starch in seeds [36]. In summary, these findings demonstrated that *FBA* is essential for metabolic processes and plant development.

In our research, *IbFBA1* and *IbFBA6* were highly expressed in storage roots, suggesting that they may be involved in starch biosynthesis of storage roots, while other *IbFBAs* were predominantly expressed in leaves and may participate in the assimilation of carbohydrates (Figure 5). As the primary harvest organ of sweet potato, storage roots serve as the main sink for carbohydrate accumulation. At different developmental stages of storage roots, *IbFBAs* exhibited differential expression patterns. In Taizhong 6, *IbFBA1*, *IbFBA4*, and *IbFBA6* were highly expressed in DR, and *IbFBA3* and *IbFBA7* were highly expressed in FR (Figure 6A). The qRT-PCR analysis showed that *IbFBA1* and *IbFBA2* were highly expressed at 60 DAP (initial thickening). *IbFBA3*, *IbFBA4*, *IbFBA6*, and *IbFBA7* were highly expressed at 75 DAP (rapid thickening and initial starch accumulation stage) (Figure 6B). Additionally, *IbFBA2*, *IbFBA3*, *IbFBA5*, and *IbFBA6* were highly expressed in the high-starch-content variety Jishu 25; *IbFBA1* and *IbFBA4* were highly expressed in the low-starch-content variety Jishu 29 (Figure 6B). The qRT-PCR analysis showed that *IbFBA2*, *IbFBA5*, and *IbFBA6* were expressed at significantly higher levels in high-starch lines compared to medium- and low-starch lines (Figure 7). Therefore, *IbFBA1* may primarily contribute to storage root development, while *IbFBA6* may be involved in both development and starch biosynthesis, making them potential targets for improving yield and starch content in sweet potato.

### 3.3. FBAs Regulate Response to Drought and Salt Stresses in Sweet Potato and Its Two Wild Diploid Relatives

*FBAs* in plants have been reported to regulate plant response to abiotic stress. In tomato, reduced *FBA* activities compromised plant tolerance to chilling stress [20]. In mini Chinese cabbage seedlings, BR enhanced S-nitrosylation of Br*FBA*2 and accelerated ATP release, helping to maintain homeostasis of cell energy metabolism at low temperature [37]. In chickpea, the activities of fructose-1,6-bisphosphate aldolase declined under drought stress [19]. In maize, the fructose-1,6-bisphosphate aldolase 1 gene (*Aldo1*) participated in the response to hypoxia [38,39]. In *Sesuvium portulacastrum*, drought and salt treatment induced the expression of *SpFBA* [18]. In this study, *IbFBA1*, *IbFBA4*, *IbFBA5*, *IbFBA6*, and *IbFBA7* in sweet potato; *ItfFBA1* and *ItfFBA6* in *I. trifida*; and *ItbFBA1*, *ItbFBA4*, and *ItbFBA6* in I. triloba were upregulated under drought stress (Figure 8A and Appendix A). Therefore, the *FBAs*, especially *FBA1* and *FBA6*, may play crucial roles in response to drought stress in sweet potato and its two diploid relatives.

Under NaCl stress, *IbFBA2*, *IbFBA4*, and *IbFBA5* were upregulated, and *IbFBA6* and *IbFBA7* were downregulated in both Jishu 26 and NL54 (Figure 8B). *ItfFBA1* and *ItfFBA6* in *I. trifida* and *ItbFBA1* and *ItbFBA6* in *I. triloba* were upregulated under NaCl stress (Appendix A). These findings suggest that *FBAs* are involved in regulating salt stress responses in sweet potato and its diploid relatives, although their salt tolerance functions may have diverged during sweet potato evolution.

### 3.4. FBAs Participate in Hormone Crosstalk in Sweet Potato and Its Two Wild Diploid Relatives

In previous studies, *FBA* genes in plants have been shown to participate in hormonal signaling pathways, including those involving abscisic acid (ABA), gibberellin (GA), and brassinosteroids (BR) [40,41]. However, no reports have addressed the roles of *FBAs* in mediating responses to other plant hormones. In this study, abundant ABA-, GA-, and SA-responsive elements were identified in the *IbFBA* promoters, and similarly, *ItfFBAs* and *ItbFBAs* promoters were enriched in hormone-responsive elements (Figure 4). Furthermore, most *IbFBAs* were transcriptionally activated by ABA, GA_3_, and IAA, whereas most *ItfFBAs* and *ItbFBAs* were repressed or unaffected by these hormone treatments (Appendix A). Specifically, *IbFBA1* and *IbFBA6*, along with their homologs, *ItfFBA1*, *ItfFBA6*, *ItbFBA1*, and *ItbFBA6*, were upregulated under ABA treatment, and *IbFBA6*, along with *ItfFBA6* and *ItbFBA6*, were upregulated by GA3 treatment (Figure 9). These results suggest that *FBAs* may participate in hormone crosstalk in sweet potato and its wild diploid relatives, and their hormonal regulatory roles may have become more pronounced during the evolution of sweet potato. *FBA1* is likely involved in ABA-mediated signaling, while *FBA6* may function in both ABA and GA responses in sweet potato and its diploid relatives.

## 4. Materials and Methods

### 4.1. Identification of FBAs

All protein sequences of *I. trifida* and *I. triloba* were extracted from the *Ipomoea* Genome Hub (https://sweetpotato.uga.edu/, accessed on 2 April 2025), and the *I. batatas* Taizhong6 protein sequences were downloaded from the Sweet potato Genomics Resource (https://sweetpotao.com/download_genome.html, accessed on 2 February 2025). The Hidden Markov Model (HMM) profiles of the glycolytic domain (PF00274) and fructose-bisphosphate aldolase class-II domain (PF01116) were downloaded from the Pfam database (http://www.ebi.ac.uk/interpro/, accessed on 2 April 2025) to screen all the possible *FBA* proteins in *I. batatas*, *I. trifida*, and *I. triloba*. All the predicted *FBAs* sequences were checked using NCBI Batch CD-Search programs (CDD, E value < 1e^−2^, https://www.ncbi.nlm.nih.gov/Structure/cdd/wrpsb.cgi, accessed on 2 April 2025) and SMART (http://smart.embl-heidelberg.de/, accessed on 2 April 2025).

### 4.2. Chromosomal Location and Property Prediction of FBAs

*IbFBAs, ItfFBAs*, and *ItbFBAs* were located on the *I. batatas*, *I. trifida*, and *I. triloba* chromosomes, respectively, according to the annotated GFF3 files downloaded from the *Ipomoea* Genome Hub (https://sweetpotao.com/download_genome.html/, accessed on 3 April 2025) and Sweet potato Genomics Resource (http://sweetpotato.plantbiology.msu.edu/, accessed on 3 April 2025). The visualization was generated by the TBtools software v1.120 (South China Agricultural University, Guangzhou, China). “One Step MCScanX Super Fast” of TBtools software v1.131 was used to obtain collinearity information among *IbFBAs, ItfFBAs* and *ItbFBAs* [42].

The ExPASy was used to calculate the MW, theoretical *p*I, unstable index, and hydrophilic of the *FBAs* (https://www.expasy.org/, accessed on 3 April 2025). The subcellular localizations of *FBAs* were presumed by WoLF PSORT website (https://wolfpsort.hgc.jp/aboutWoLF_PSORT.html.en, accessed on 3 April 2025).

### 4.3. Phylogenetic Analysis of FBAs

Protein sequences of *A. thaliana*, *I. batatas*, *I. triloba*, *I. trifida*, and *S. tuberosum* were used to construct the phylogenetic tree via the neighbor-joining method with 1000 bootstrap replicates [43]. The phylogenetic tree was visualized by iTOL (http://itol.embl.de/, accessed on 5 April 2025).

### 4.4. Gene Structures and Conserved Motifs Analysis of FBAs

The conserved motifs of *FBAs* were analyzed using MEME software (https://meme-suite.org/meme/tools/meme, accessed on 5 April 2025), and the maximum number of motif parameters was set to 10 [44]. Tbtools software (v1.131) was used to visualize the conserved domain structures and exon–intron structures of *FBAs* (South China Agricultural University, Guangzhou, China).

### 4.5. Cis-Acting Elements in Promoter Regions Analysis of IbFBAs

The PlantCARE was used to predicted the *cis*-elements in the approximately 2000 bp promoter regions of *FBAs* (http://bioinformatics.psb.ugent.be/webtools/plantcare/html/, accessed on 5 April 2025) [45]. TBtools software (v1.131) was used for visualization.

### 4.6. Expression Patterns Analysis of IbFBAs

Transcriptome data of Taizhong 6 in different developmental stages (FR, fibrous roots; DR, developing storage roots; MR, mature storage roots) and storage roots expansion of Jishu 29 and Jishu 25 under different developmental stages (32, 46 and 67 DAP: day after planting) from a previous study with NCBI project ID PRJNA756699 were used to calculate the expression patterns of *IbFBAs* [30,32]. In addition, transcriptome data of the salt-tolerant variety Jishu 26 and the salt-sensitive variety NL54 under salt stress were obtained from previous studies under NCBI project number PRJNA552932 [31]. The RNA-seq data of *FBAs* in *I. trifida* and *I. triloba* were downloaded from the Sweetpotato Genomics Resource (http://sweetpotato.plantbiology.msu.edu/, accessed on 4 March 2023). The expression levels of *FBAs* were calculated as fragments per kilobase of exon per million fragments mapped (FPKM), and the heat maps of expression were constructed using TBtools software v1.120 (South China Agricultural University, Guangzhou, China).

For the analysis of *IbFBAs* expression in different organs, under stress, and with hormone effects, the seedlings of sweet potato variety ‘Jishu25’ were collected from the Crop Research Institute, Shandong Academy of Agricultural Sciences, China. Seedlings were grown in Hoagland solution under a light cycle of 26 °C, 16 h of illumination, and 8 h of darkness. When seedlings have 5 to 6 functional leaves and 8 to 10 cm of adventitious roots, they are subjected to four different treatments. The storage roots of Jishu25 sampled at the different development stages (30, 45, 60, 75, 90, 105, and 120 DAP) were used for analyzing the expression of *IbFBAs*. Hoagland’s solution containing 20% PEG 6000, 100 mmol/L IAA, 100 mmol/L ABA, and 100 mmol/L GA3 was used, respectively,; treated fibrous roots were collected after 0, 3, 6, 12, 24, and 48 h. Total RNA was extracted from the sample using an RNA isolator, Total RNA Extraction Reagent (Vzayme, Nanjing, China). cDNA was obtained through reverse transcription using a reverse transcription kit (Takara, Beijing, China) and used as a template. The CFX Connect real-time system (Bio-RAD) and ChamQ Universal SYBR qPCR Master Mix (Vazyme, Nanjing, China) were used for qRT-PCR. Using the *Ibactin* gene as an internal reference, Appendix A lists the primer sequences of the examined genes. The experiment was repeated three times, and the data were calculated using the 2^−△△CT^ method [46].

### 4.7. Statistical Analysis

All data were analyzed using a two-tailed Student’s *t*-test with SPSS 26.0 (https://www.ibm.com/support/pages/downloading-ibm-spss-statistics-26, accessed on 14 March 2025). Data are shown as means ± standard deviation (SD). Heatmaps were generated using TBtools software v1.131.

## 5. Conclusions

Seven, seven, and six *FBAs* were identified in the cultivated sweet potato and its two wild diploid relatives *I. trifida* and *I. triloba*, respectively. A comprehensive analysis was performed on their protein physicochemical properties, chromosomal localization, phylogenetic relationship, gene structure, promoter *cis*-elements, protein interaction network, and expression patterns. All the *IbFBAs, ItfFBAs*, and *ItbFBAs* shared the same conserved motifs in proteins, and *IbFBAs* had the same exons structures as their homologs (*ItfFBAs* and *ItbFBAs*). The conserved gene and protein structures suggest functional conservation among *FBAs*. The *FBAs* exhibited differential expressions across various organs. *IbFBAs* might play vital roles in storage root development and starch biosynthesis in sweet potato; among them, *IbFBA1* and *IbFBA6* are promising candidate genes for further functional investigation. In addition, the *FBAs* were responsive to drought and salt stress. They also took part in hormone crosstalk, especially between ABA and GA. This work offers valuable insights into the structure and function of *FBAs* and identifies candidate genes for enhancing yield, starch content, and abiotic stress tolerance in sweet potatoes. By leveraging the genetic diversity and functional plasticity of *FBA* genes, we can develop crops with enhanced photosynthetic efficiency and resilience to environmental stresses, contributing to global food security.

## Figures and Tables

**Figure 1 ijms-26-07348-f001:**
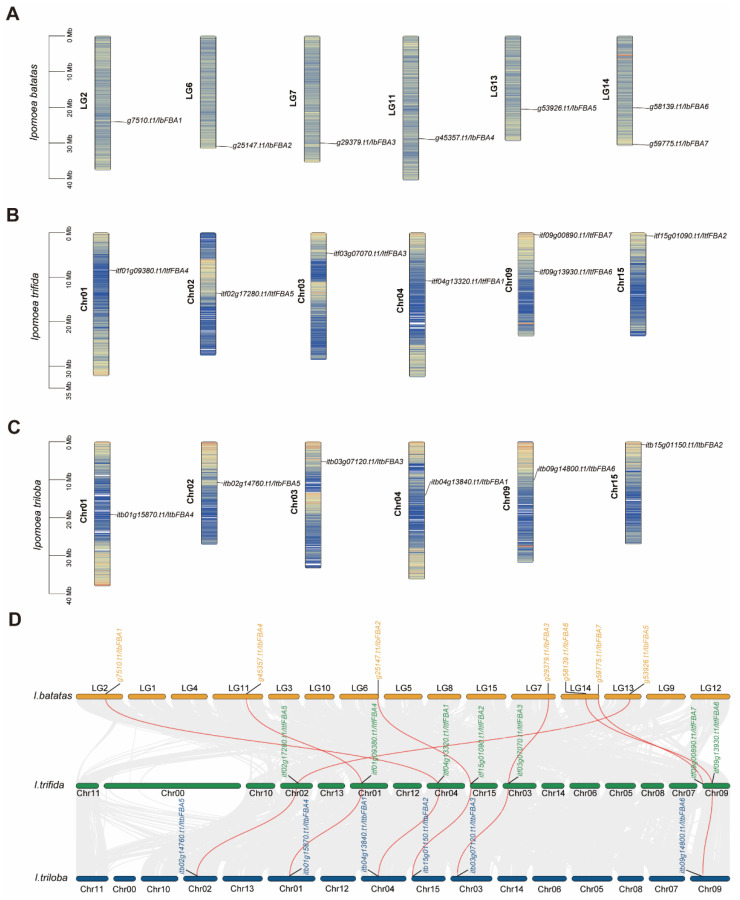
Chromosomal localization and distribution of *FBAs* in *I. batatas* (**A**), *I. trifida* (**B**), and *I. triloba* (**C**). The bars represent chromosomes. The chromosome numbers are displayed on the left side, and the gene names are displayed on the right side. The relative chromosomal localization of each *FBA* gene is marked on the black line of the right side and indicated by the unit Mbp. The stripes on chromosomes represent the density of chromosomes. (**D**) Syntenic analysis of *I. batatas*, *I. trifida*, and *I. triloba FBAs*. Chromosomes of *I. batatas*, *I. trifida*, and *I. triloba* are shown in different colors. The approximate positions of *IbFBAs*, *ItfFBAs*, and *ItbFBAs* are marked with short black lines on the chromosomes. Red curves denote the syntenic relationships between *I. batatas* and *I. trifida FBAs*.

**Figure 2 ijms-26-07348-f002:**
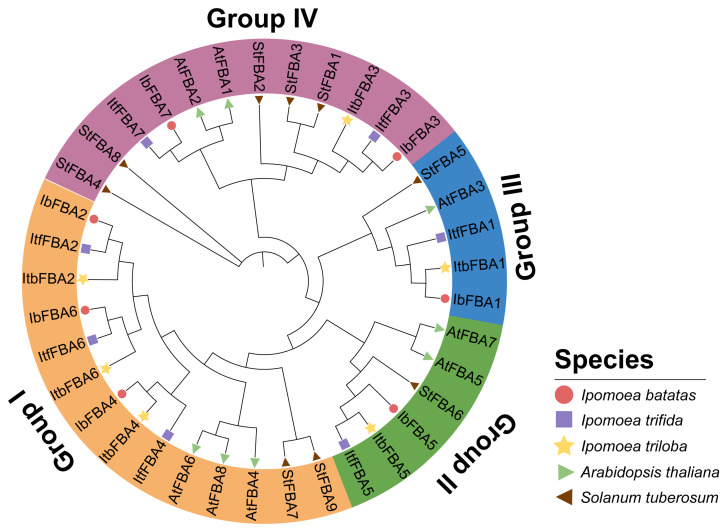
Phylogenetic analysis of *FBAs* in *I. batatas*, *I.triloba*, *I. trifida*, *A. thaliana*, and *S. tuberosum*. Based on the neighbor-joining method with 1000 bootstrap replicates, a total of 37 *FBAs* were divided into four groups (groups I, II, III, and IV, filled with orange, green, blue, and purple, respectively). The claret circles represent seven Ib*FBAs* in *I. batatas*. The purple squares represent seven Itf*FBAs* in *I. trifida*. The yellow stars represent six Itb*FBAs* in *I. triloba*. The green triangles represent eight At*FBAs* in *A. thaliana*. The brown triangles represent nine St*FBAs* in *S. tuberosum*.

**Figure 3 ijms-26-07348-f003:**
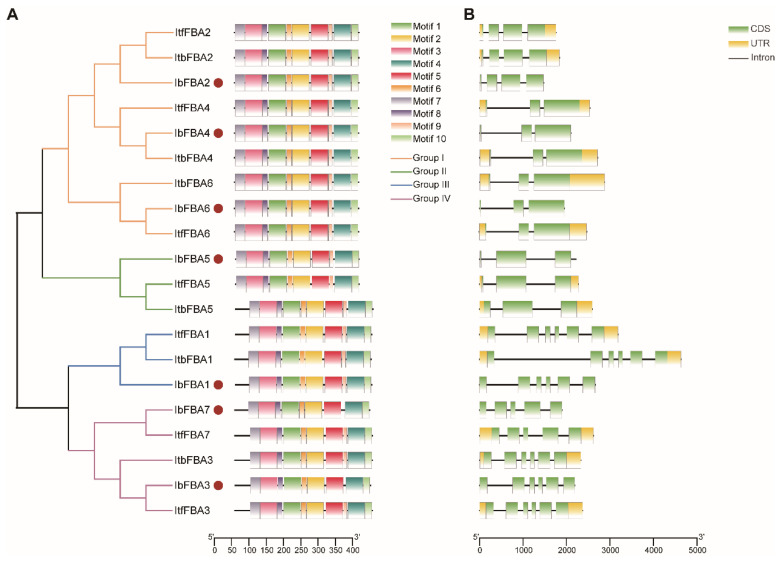
Conserved motifs and exon–intron structure analysis of *FBAs* in *I. batatas*, *I. trifida*, and *I. triloba*. (**A**) The phylogenetic tree showed that *FBAs* were divided into four subgroups, and the ten conserved motifs were shown in different colors. The claret circles represent Ib*FBAs*. (**B**) Exon–intron structures of *FBAs*. The green boxes, yellow boxes, and black lines represent CDS, UTRs, and introns, respectively.

**Figure 4 ijms-26-07348-f004:**
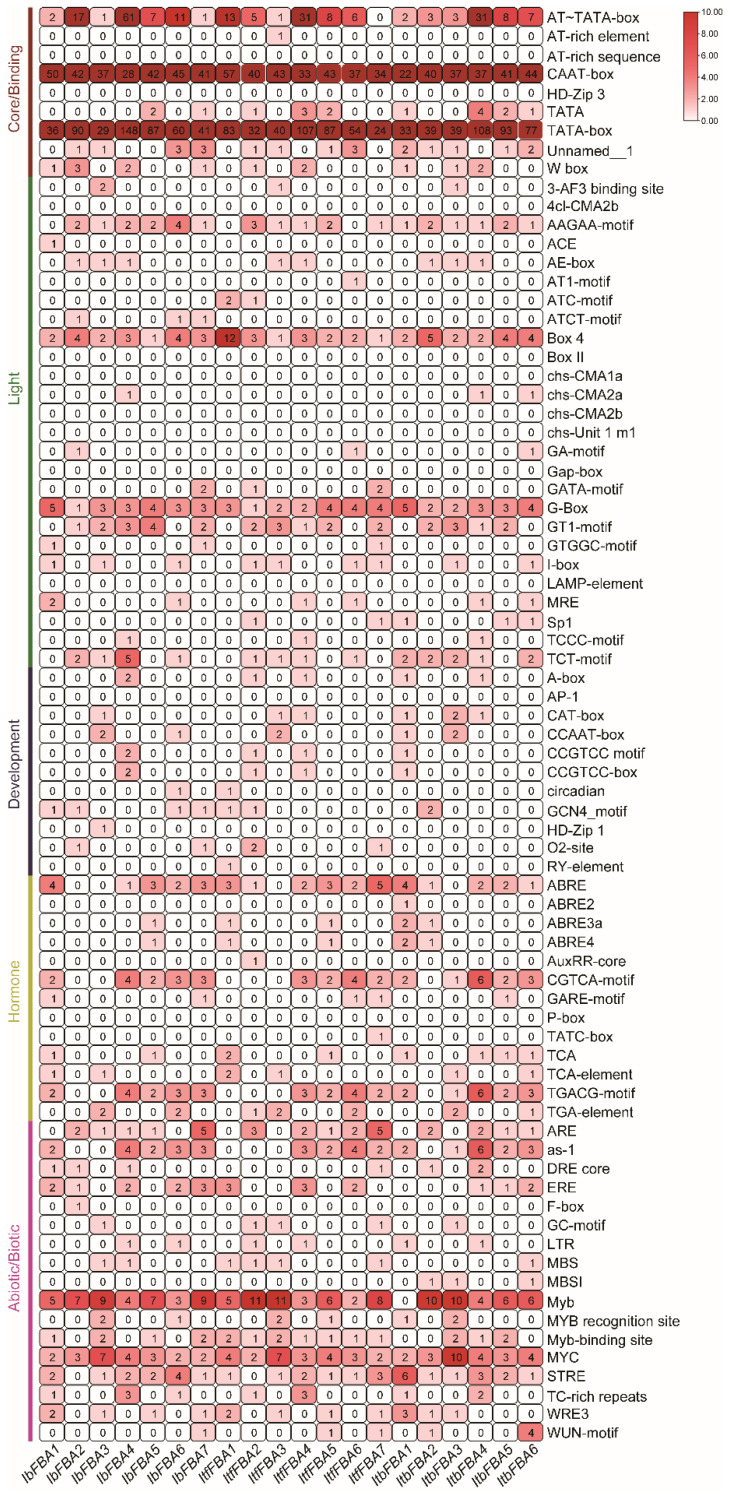
*Cis*-element analysis in the promoters of *FBAs* from *I. batatas*, *I. trifida*, and *I. triloba*. The *cis*-elements were divided into five broad categories. The degree of red colors represents the number of *cis*-elements in the promoters of *FBAs*.

**Figure 5 ijms-26-07348-f005:**
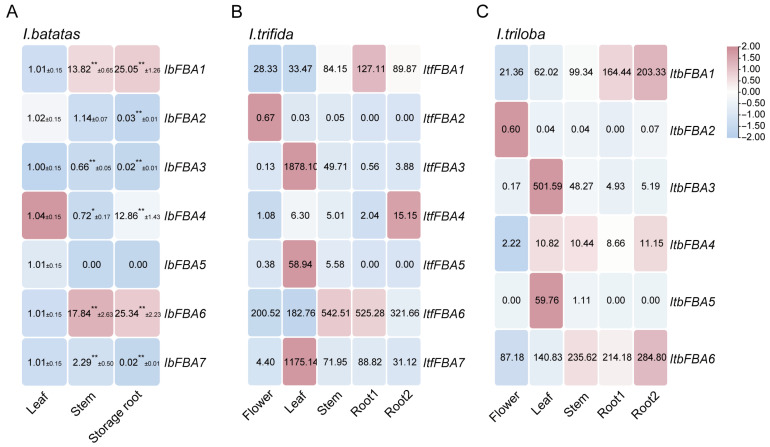
Expression analysis of *FBAs* in different organs of *I. batatas*, *I. trifida*, and *I. triloba*. (**A**) Expression analysis of *IbFBAs* in the leaf, stem, and storage root of *I. batatas*. The values were determined by qRT-PCR from three biological replicates consisting of pools of three plants, and the results were analyzed using the comparative C_T_ method. The expression level of the leaf is determined as “1”. Fold change ± SD is shown in the boxes. (**) and (*) indicate significant difference at *p* < 0.01 and *p* < 0.05, respectively, based on the Student’s *t-*test compared to the leaf. (**B**) Expression patterns of *ItfFBAs* in the flower, leaf, stem, root1, and root2 of *I. trifida*. (**C**) Expression patterns of *ItbFBAs* in the flower, leaf, stem, root1, and root2 of *I. triloba*. The FPKM values are shown in the boxes.

**Figure 6 ijms-26-07348-f006:**
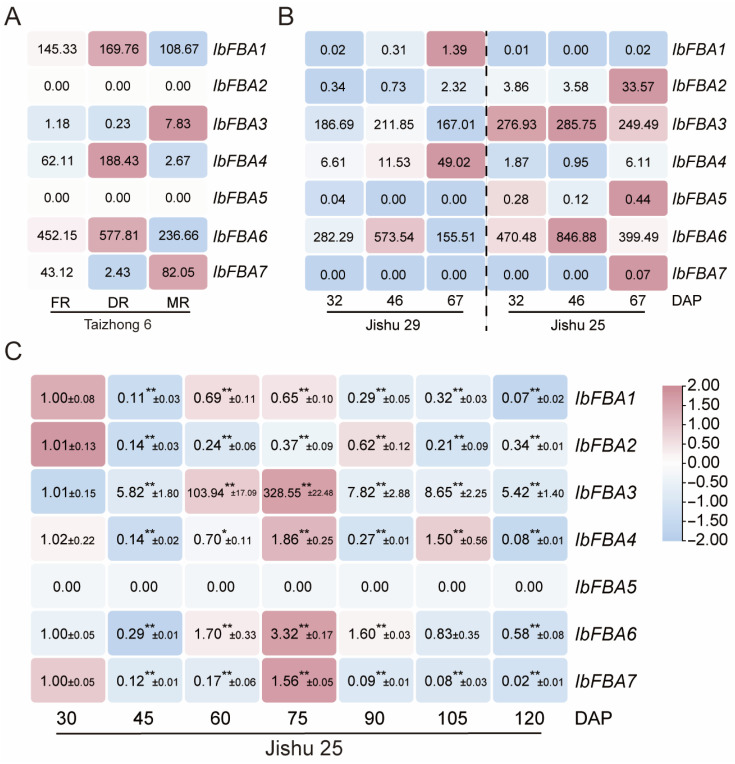
Expression analysis of *IbFBAs* at different developmental stages of the H283 storage roots (i.e., 20, 30, 40, 50, 60, 70, 80, 90, 100, and 130 DAP) using qRT-PCR. (**A**) Expression analysis of *IbFBAs* at different developmental stages of the Taizhong 6 storage roots (i.e., FR, DR, and MR). The FPKM values are shown in the boxes. (**B**) Expression analysis of *IbFBAs* at different developmental stages of the Jishu 29 and Jishu 25 storage roots (i.e., 32, 46, and 67 DAP). The FPKM values are shown in the boxes. (**C**) Expression analysis of *IbFBAs* at different developmental stages of the Jishu 25 storage roots (i.e., 30, 45, 60, 75, 90, 105, and 120 DAP) using qRT-PCR. The values were determined via qRT-PCR from three biological replicates consisting of pools of three plants, and the results were analyzed using the comparative C_T_ method. The expression level of 30 DAP is determined as “1”. Fold change ± SD is shown in the boxes. (**) and (*) indicate significant difference at *p* < 0.01 and *p* < 0.05, respectively, based on the Student’s *t-*test compared to 30 DAP.

**Figure 7 ijms-26-07348-f007:**
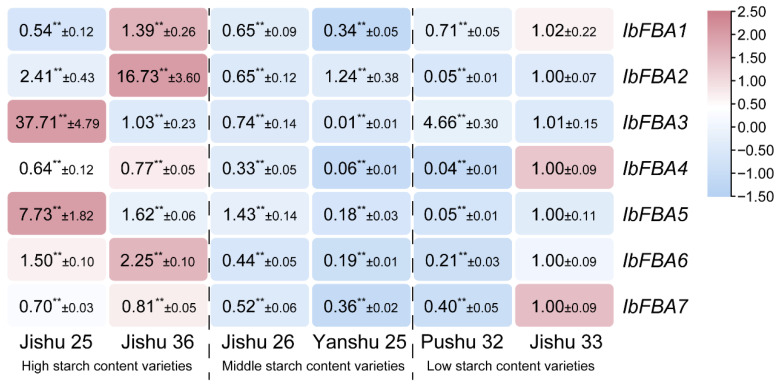
Expression analysis of *IbFBAs* in sweet potato lines with different starch contents. The values were determined using qRT-PCR from three biological replicates consisting of pools of three plants, and the results were analyzed using the comparative C_T_ method. The expression level of Jishu 33 is determined as “1”. Fold change ± SD is shown in the boxes. (**)indicates significant difference at *p* < 0.01 based on the Student’s *t-*test compared to Jishu 33.

**Figure 8 ijms-26-07348-f008:**
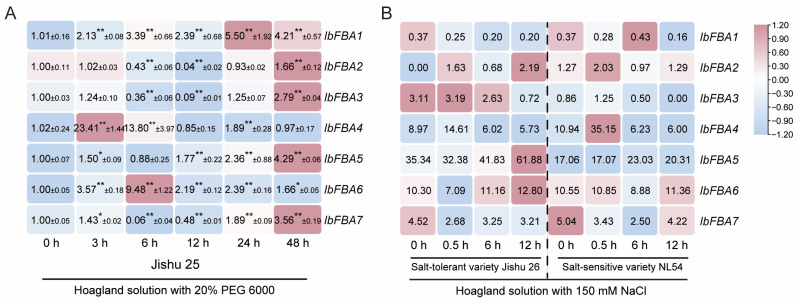
Expression analysis of *IbFBAs* in sweet potato under drought and salt stresses. (**A**) Expression analysis of *IbFBAs* in the drought-tolerant line Xu55-2 under PEG6000 stress. The values were determined with the qRT-PCR from three biological replicates consisting of pools of three plants, and the results were analyzed using the comparative C_T_ method. The expression level of 0 h is determined as “1”. Fold change ± SD is shown in the boxes. (**) and (*) indicate significant difference at *p* < 0.01 and *p* < 0.05, respectively, based on the Student’s *t-*test compared to 0 h. (**B**) Expression analysis of *IbFBAs* in the salt-sensitive variety NL54 and salt-tolerant line Jishu 26 under NaCl stress. The FPKM values are shown in the boxes.

**Figure 9 ijms-26-07348-f009:**
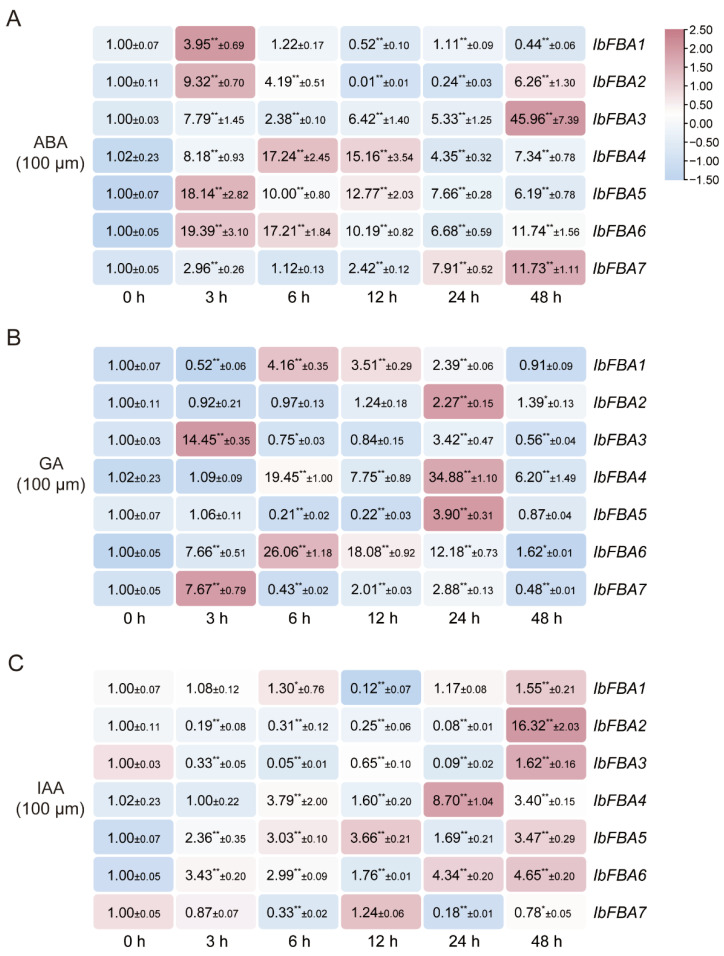
Expression analysis of *IbFBAs* in response to different hormones in sweet potato line Jishu 25: (**A**) ABA. (**B**) GA3. (**C**) IAA. The values were determined using the qRT-PCR from three biological replicates consisting of pools of three plants, and the results were analyzed using the comparative C_T_ method. The expression level of 0 h is determined as “1”. Fold change ± SD is shown in the boxes. (**) and (*) indicate significant difference at *p* < 0.01 and *p* < 0.05, respectively, based on the Student’s *t-*test compared to 0 h.

**Table 1 ijms-26-07348-t001:** Characteristics of *FBAs* in *I. batatas*, *I. trifida*, and *I. triloba*.

Gene ID	Gene Name	Genomic Length (bp)	CDS (bp)	Protein Size (aa)	MW (kDa)	*p*I	Instability	GRAVY	Subcellular Location
g7510.t1	*IbFBA1*	2902	1182	393	42.49	8.61	38.95	−0.238	chloroplast
g25147.t1	*IbFBA2*	1682	1080	359	38.97	7.53	32.85	−0.336	cytoplasm
g29379.t1	*IbFBA3*	2345	1173	390	41.91	7.56	35.57	−0.087	chloroplast
g45357.t1	*IbFBA4*	2537	1077	358	38.62	7.51	34.46	−0.228	cytoplasm
g53926.t1	*IbFBA5*	2702	1047	348	37.14	6.86	36.14	−0.33	cytoplasm
g58139.t1	*IbFBA6*	2418	1077	358	38.51	7.53	29.23	−0.156	cytoplasm
g59775.t1	*IbFBA7*	2134	1170	389	41.41	8.17	36.61	−0.13	chloroplast
itf15g01090	*ItfFBA1*	1752	1080	359	38.97	7.53	32.85	−0.336	cytoplasm
itf09g13930	*ItfFBA2*	2461	1077	358	38.51	7.53	29.23	−0.156	cytoplasm
itf03g07070	*ItfFBA3*	2365	1194	397	42.66	7.59	36.02	−0.105	chloroplast
itf04g13320	*ItfFBA4*	3179	1182	393	42.49	8.61	38.95	−0.238	chloroplast
itf02g17280	*ItfFBA5*	2273	1074	357	38.23	6.38	31.82	−0.123	cytoplasm
itf01g09380	*ItfFBA6*	2527	1077	358	38.61	7.51	34.09	−0.228	cytoplasm
itf09g00890	*ItfFBA7*	2611	1194	397	42.58	8.49	36.56	−0.094	chloroplast
itb04g138401	*ItbFBA1*	4627	1182	393	42.49	8.61	38.95	−0.238	chloroplast
itb15g011501	*ItbFBA2*	1831	1080	359	38.97	7.53	32.85	−0.336	cytoplasm
itb01g158701	*ItbFBA3*	2699	1077	358	38.62	8	35.12	−0.228	cytoplasm
itb09g148001	*ItbFBA4*	2864	1077	358	38.51	8.04	30.13	−0.156	cytoplasm
itb03g071201	*ItbFBA5*	2316	1194	397	42.67	7.59	36.02	−0.101	chloroplast
itb02g147601	*ItbFBA6*	2589	1191	396	42.69	6.97	38.86	−0.101	chloroplast

CDS, coding sequence; MW, molecular weight; *p*I, isoelectric point; GRAVY, grand average of hydropathicity.

## Data Availability

The original contributions presented in this study are included in the article/Appendix A. Further inquiries can be directed to the corresponding author.

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
