# Peer review of "Genome-Wide Identification and Expression Analysis of the Fructose-1,6-Bisphosphate Aldolase (FBA) Gene Family in Sweet Potato and Its Two Diploid Relatives"

_ijms, 2025, doi:10.3390/ijms26157348_

Round 1
Reviewer 1 Report
Comments and Suggestions for Authors
The manuscript was well characterized on 20 FBAs identified from cultivated sweetpotato and its two diploid wild relatives in terms of their protein physicochemical properties, chromosomal localization, phylogenetic relationship etc. Some FABs genes can be useful genes in improving yield, starch content and abiotic stress tolerance for molecular breeding of sweetpotato. Thus I would like to recommend the manuscript for the publication after a minor revision.
The scientific name should be in italic in all text and references (Line 57 Nicotiana tabacum, Table 1. Characteristics of FBAs in I. batatas, I. trifida, and I. triloba., Line 547 Galdieria sulphuraria, Line 557 Arabidopsis, Line 560 Camellia oleifera).
At reference 36, the tile should be a small character like “Unveiling the biological function of phyllostachys ----”
In addition, I would like to recommend to unify sweet potato as sweetpoato. CIP and Sweetpotato Institute in Chinese Academy of Agricultural Sciences use sweetpotato as a official word.
Author Response
Reviewer 1:
The manuscript was well characterized on 20 FBAs identified from cultivated sweetpotato and its two diploid wild relatives in terms of their protein physicochemical properties, chromosomal localization, phylogenetic relationship etc. Some FABs genes can be useful genes in improving yield, starch content and abiotic stress tolerance for molecular breeding of sweetpotato. Thus I would like to recommend the manuscript for the publication after a minor revision.
Response:
Thank you very much for your affirmation of our research and article.
- The scientific name should be in italic in all text and references (Line 57 Nicotiana tabacum, Table 1. Characteristics of FBAs in I. batatas, I. trifida, and I. triloba., Line 547 Galdieria sulphuraria, Line 557 Arabidopsis, Line 560 Camellia oleifera).
Response:
We thank the reviewer for this comment. Based on the comment, we have made careful modifications.
- At reference 36, the tile should be a small character like “Unveiling the biological function of phyllostachys ----”
Response:
We thank the reviewer for this comment. Based on the comment, we have made careful modifications.
- In addition, I would like to recommend to unify sweet potato as sweetpoato. CIP and Sweetpotato Institute in Chinese Academy of Agricultural Sciences use sweetpotato as a official word.
Response:
We thank the reviewer for pointing this out. We have standardized the term to ‘sweetpotato’ throughout the revised manuscrip.
Reviewer 2 Report
Comments and Suggestions for Authors
Review on “Genome-Wide Identification and Expression Analysis of the Fructose-1,6-Bisphosphate Aldolase (FBA) Gene Family in Sweet Potato and Its Two Diploid Relatives” for manuscript ID ijms-3748872
In this manuscript the authors describe the FBA gene family encoding key glycolysis and Calvin cycle enzyme in sweet potato and its wild relatives. Despite knowledge in other plants, the identification and functional analysis of FBA genes in sweet potato are lacking. This study aims to fill that gap by systematically analyzing the FBA gene family within sweet potato and its wild relatives, focusing on their structures, expression patterns, and potential roles in development and stress responses. The work is valuable since it provides novel data on structure and function of FBAs that could be used for crops improvement.
Major points:
Fig. 1 A-C: It is not stated in the figure caption what do the stripes on chromosomes represent. Gene density?
Fig. 2: You need to clarify the method used for phylogeny reconstruction in the figure caption.
Fig. 3 B: The colors stated in the figure caption do not match with ones on the figure.
L158: SUSs??
Minor points:
Fig. 5A: The significance level asterisks are very small and hard to read. It would be better to increase the visibility of significance level - maybe make asterisks bigger, or use high-contrast color.
Fig. 6C and Fig.8A: same as above.
L86: all species names would be italicized
Figure 2: Group II, III names written incorrectly
Author Response
Reviewer 2:
Review on “Genome-Wide Identification and Expression Analysis of the Fructose-1,6-Bisphosphate Aldolase (FBA) Gene Family in Sweet Potato and Its Two Diploid Relatives” for manuscript ID ijms-3748872
In this manuscript the authors describe the FBA gene family encoding key glycolysis and Calvin cycle enzyme in sweet potato and its wild relatives. Despite knowledge in other plants, the identification and functional analysis of FBA genes in sweet potato are lacking. This study aims to fill that gap by systematically analyzing the FBA gene family within sweet potato and its wild relatives, focusing on their structures, expression patterns, and potential roles in development and stress responses. The work is valuable since it provides novel data on structure and function of FBAs that could be used for crops improvement.
Major points:
- 1 A-C: It is not stated in the figure caption what do the stripes on chromosomes represent. Gene density?
Response:
We thank the reviewer for the suggestion. As suggested, we added detailed information to explain what the stripes on chromosomes represent: “The stripes on chromosomes the density of chromosomes.” [Lines 118-119].
- 2: You need to clarify the method used for phylogeny reconstruction in the figure caption.
Response:
We thank the reviewer for the suggestion. As suggested, we added detailed information to clarify the method used for phylogeny reconstruction: “Based on the neighbor-joining method with 1000 bootstrap replicates, a total of 37 FBAs were divided into four groups (groups I, II, III and IV, filled with orange, green, blue and purple, respectively).” [Lines 136-138].
- 3 B: The colors stated in the figure caption do not match with ones on the figure.
Response:
We thank the reviewer for pointing this out. As suggested, we carefully checked the colors stated in the figure caption with Fig. 3 to match with the figure: “The claret circles represent IbFBAs.” [Line 154] and “The green boxes, yellow boxes, and black lines represent CDS, UTRs and introns, respectively.” [Lines 155-156].
- L158: SUSs??
Response:
We thank the reviewer for pointing this out. We have corrected the term to ‘FBAs’ [Line 157].
Minor points:
- 5A: The significance level asterisks are very small and hard to read. It would be better to increase the visibility of significance level - maybe make asterisks bigger, or use high-contrast color.
Response:
Thank you very much for your suggestions. Based on the comments, we have made asterisks bigger in the revised manuscript.
- 6C and Fig.8A: same as above.
Response:
Thank you very much for your suggestions. Based on the comments, we have made asterisks bigger in the revised manuscript.
- L86: all species names would be italicized
Response:
We thank the reviewer for this comment. Based on the comment, we have made careful modifications.
- Figure 2: Group II, III names written incorrectly
Response:
We thank the reviewer for this comment. Based on the comment, we have made careful modifications.

Reviewer 3 Report
Comments and Suggestions for Authors
This paper presents a bioinformatic analysis of the gene family of enzyme of Fructose-1,6-bisphosphate aldolase in sweet potato genome.
Like other numerous similar works, this work has some novelties. Specifically, the analysis of 3 species of Ipomoea batatas, I. trifida and six in I. triloba.
The gene encoding this enzyme is located exclusively in the chloroplast genome, however, the authors search for it found in the nuclear genome where it does not exist.
The second problem, the authors write "their tissue specificity and expression patterns related to storage root development, starch biosynthesis and abiotic stress and hormone responses were examined using qRT-PCR or RNA-seq."
It is a characteristic feature of such works to give purely virtual work, features of applied work. However, in this case it was only necessary to analyse the enzyme activity on stress and determine the amount of enzyme. No qRT-PCR or RNA-seq is needed in this work. If enzymes are analysed, then enzymes should be analysed as well.
All methods and pictures in this paper are not different from other similar papers on other genes and their virtual analysis. Authors need to provide data for each gene in the form of NCBI accesion ID to be able to prove the presence of results in the work.
Sequences of primers, as I wrote above, it is not required, and secondly, it is necessary to have more data on these primers.
Author Response
Reviewer 3:
This paper presents a bioinformatic analysis of the gene family of enzyme of Fructose-1,6-bisphosphate aldolase in sweet potato genome.
Like other numerous similar works, this work has some novelties. Specifically, the analysis of 3 species of Ipomoea batatas, I. trifida and six in I. triloba.
- The gene encoding this enzyme is located exclusively in the chloroplast genome, however, the authors search for it found in the nuclear genome where it does not exist.
Response:
We thank the reviewer for this comment. Actually, FBA is a key enzyme that catalyzes the reversible conversion of fructose-1,6-bisphosphate (FBP) into dihydroxyacetone phosphate (DHAP) and glyceraldehyde-3-phosphate (G3P) and plays distinct roles in glycolysis and Calvin cycle. Therefore, in higher plants, FBA is localized not only in the chloroplasts but also in cytoplasm, which indicated that FBA genes exist in nuclear genome. Moreover, recent studies have shown that FBA genes exist in the nuclear genomes of multiple species, including eight in Arabidopsis thaliana, eight in Solanum lycopersicum, 16 in Nicotiana tabacum, 17 in cotton, nine in Solanum tuberosum, and five in Cucumis sativus. In summary, FBA genes are widely present in the nuclear genomes of various plants, and studying FBA genes in sweetpotato and its two diploid wild relatives nuclear genomes is very meaningful.
- The second problem, the authors write "their tissue specificity and expression patterns related to storage root development, starch biosynthesis and abiotic stress and hormone responses were examined using qRT-PCR or RNA-seq." It is a characteristic feature of such works to give purely virtual work, features of applied work. However, in this case it was only necessary to analyse the enzyme activity on stress and determine the amount of enzyme. No qRT-PCR or RNA-seq is needed in this work. If enzymes are analysed, then enzymes should be analysed as well.
Response:
We thank the reviewer for this comment. Our research aims to provide candidate genes for improving sweetpotato yield, quality and stress tolerance by studying the roles of all FBA genes in sweetpotato and its diploid wild species, rather than investigating the relationship between FBA enzyme activity and storage root development, starch biosynthesis and abiotic stress and hormone responses. Therefore, we need qRT-PCR or RNA-seq to explore the roles of all FBA genes in storage root development, starch biosynthesis and abiotic stress and hormone responses, and the determination and quantitative study of FBA enzyme activity cannot meet our demand for candidate genes in sweetpotatoes. Moreover, the transcription level of genes can to some extent reflect the activity and quantity of enzymes, so it is correct and feasible to use qRT-PCR or RNA-seq for research in this study.
- All methods and pictures in this paper are not different from other similar papers on other genes and their virtual analysis. Authors need to provide data for each gene in the form of NCBI accession ID to be able to prove the presence of results in the work.
Response:
Thank you very much for your suggestions. As suggestion, we provide data for each gene in the form of NCBI accession ID in Table S1.
- Sequences of primers, as I wrote above, it is not required, and secondly, it is necessary to have more data on these primers.
Response:
Thank you very much for your suggestions. According to the suggestion, we added the GC content and annealing temperature of primers in Table S2.

Reviewer 4 Report
Comments and Suggestions for Authors
The work studied the whole-genome identification and expression analysis of the fructose-1,6-bisphosphate aldolase (FBA) family genes in sweet potato and two of its diploid relatives. Some comments.
It is desirable to provide a photo of the studied plant.
In the Materials and Methods section, it is necessary to provide a separate subsection on statistical analysis.
In subsection 2.5 in the Results section, it is necessary to talk about organs, not tissues.
FBA genes are involved in responses to various stresses, such as salinity, drought, and temperature, but it is also necessary to determine them 24 h after the stress.
It is also desirable to determine the ROS content under and after stress.
It is also desirable to determine the starch content by histology.
It is desirable to determine the ploidy of wild and cultivated plants under stress.
It is desirable to provide microphotographs of the root system.
Author Response
Reviewer 4:
The work studied the whole-genome identification and expression analysis of the fructose-1,6-bisphosphate aldolase (FBA) family genes in sweet potato and two of its diploid relatives. Some comments.
- It is desirable to provide a photo of the studied plant.
Response:
We thank the reviewer for the suggestion. As suggested, we provide a photo of morphology and characteristics of sweetpotato storage roots (SRs) at different stages in two cultivars according to Du et al (2023).
Referance:
Du, T. F.; Qin, Z.; Zhou, Y. Y.; Zhang, L. M.; Wang, Q. M.; Li, Z. Y.; Hou, F. Y. Comparative transcriptome analysis reveals the effect of lignin on storage roots formation in two sweetpotato (Ipomoea batatas (L.) lam.) cultivars. Genes 2023, 14, 1263.
- In the Materials and Methods section, it is necessary to provide a separate subsection on statistical analysis.
Response:
We thank the reviewer for the suggestion. As suggested, we added separate subsection on statistical analysis in the Materials and Methods section: “All data were analyzed using a two-tailed Student’s t-test with SPSS 26.0 (https://www.ibm.com/support/pages/downloading-ibm-spss-statistics-26). Data are shown as means ± standard deviation (SD). Heatmaps were generated using TBtools software v1.131 .” [Lines 470-474].
- In subsection 2.5 in the Results section, it is necessary to talk about organs, not tissues.
Response:
We thank the reviewer for pointing this out. We have standardized the term to ‘organs’ throughout the revised manuscrip.
- FBA genes are involved in responses to various stresses, such as salinity, drought, and temperature, but it is also necessary to determine them 24 h after the stress.
Response:
We thank the reviewer for the suggestion. In fact, as shown in the Figure 8A and 9, we measured the expression level of FBA genes at 48 h after treatment.
- It is also desirable to determine the ROS content under and after stress.
Response:
We thank the reviewer for the suggestion. In fact, we have measured physiological indicators such as ROS content that can reflect the degree of stress on sweetpotatoes. But these data will be applied to another achievement, which is not convenient to display in this manuscript.
- It is also desirable to determine the starch content by histology.
Response:
We thank the reviewer for the suggestion. In fact, six varieties with different starch contents were used in the manuscript, and their starch contents have been widely studied. The relevant references for the research are listed below. Jishu 36 and Jishu 33 are newly developed by Crop Research Institute of Shandong Academy of Agricultural Sciences. We have measured the starch content of these two varieties. But these data will be applied to another achievement, which is not convenient to display in this manuscript. Based on previous research results and our measurement results, six varieties were classified into high, medium, and low starch content
Referance:
Mao, S. S.; Pei, Z. C.; Yang, L. G.; Li, Z. Q.; Cao, Y.; Sun, D.; Li, R. K. Quality analysis of 12 sweet potato varieties planted in Beijing. Bull. Agric. Sci. Tech. 2024, 7, 61-64.
- It is desirable to determine the ploidy of wild and cultivated plants under stress.
Response:
We thank the reviewer for the suggestion. Sweetpotato (Ipomoea batatas (L.) Lam., 2n = B1B1B2B2B2B2 = 6x = 90) is classified as an autohexaploid species within the Convolvulaceae family, Ipomoea Genus, and Batatas Section and Its two diploid wild relatives I. trifida and I. triloba are classified as diploid species within the Convolvulaceae family, Ipomoea Genus.
- It is desirable to provide microphotographs of the root system.
Response:
We thank the reviewer for the suggestion. As suggested, we provide a photo of the occurrence, proliferation and growth of amyloplasts in sweetpotato storage roots according to Jing et al (2013).
Referance:
Jing, Y. P.; Li, D. L.; Liu, D. T.; Yu, X. R.; Hu, M. L.; Gu, Y. J.; Wang, Z. Anatomical structure of the tuberous root growth and its amyloplast development in sweet potato. Acta Bot. Boreal. Occident. Sin. 2013, 33, 2415-2422.

Round 2
Reviewer 3 Report
Comments and Suggestions for Authors
The presence of the chloroplast Fructose-1,6-bisphosphate aldolase gene in nuclear DNA is not proof that this gene is functional. These may be pseudogenes or other similar sequences.
The authors should provide the sequences of all genes identified in the nuclear genome and perform multiple alignment of DNA sequences and attach them as a supplement.
Exons and introns should be indicated in the alignment.
Show where the primers were selected for which sequences and with which software this was carried out.
The information provided by the authors in Tables S2 - ‘GC content (F/R) / Annealing Temperature (F/R)’ is of novaluable. I did not ask for this information, but for information that may be useful to someone. These are coordinates in genes, introns or exons, on what basis these primers were selected, and the size of the PCR products. Data on in silico PCR analysis for each pair of primers in this genome.
For In silico PCR analysis, how to use:
Kalendar R, Shevtsov A, Otarbay Z, Ismailova A 2024. In silico PCR analysis: a comprehensive bioinformatics tool for enhancing nucleic acid amplification assays. Frontiers in Bioinformatics, 4: 1464197. DOI:10.3389/fbinf.2024.1464197
https://www.frontiersin.org/journals/bioinformatics/articles/10.3389/fbinf.2024.1464197/full
Whether the authors like it or not, the work is dedicated to isozymes, so it is imperative to conduct a enzyme analysis of each line and stress response.
qPCR analysis as such, and in the form presented by the authors, has no value or scientific validity.
The authors simply reproduce an article similar to others, which is the only reason they use this method.
qPCR analysis is a professional method that requires not only information on each pair of primers, but also individual analysis of each pair for different mRNA regions. Only in the form of multiplex qPCR does this analysis make any sense.
These are some of my arguments as to why this work should not be published anywhere.
Second, and most importantly, the genes in the nuclear DNA of this enzyme are unconfirmed information. The authors need to do as I indicated above, collect the DNA sequences of all genes in the nuclear DNA, align them, and indicate the introns and exons.
Only then can I verify for myself whether this is correct or not.
However, since the paper only contains graphical illustrations, as in other similar articles, and lacks any data for verification, the paper cannot be published.
Unfortunately, many similar articles have been published.